# Biogeography and Genetic Diversity of Terrestrial Mites in the Ross Sea Region, Antarctica

**DOI:** 10.3390/genes14030606

**Published:** 2023-02-28

**Authors:** Gemma E. Collins, Monica R. Young, Peter Convey, Steven L. Chown, S. Craig Cary, Byron J. Adams, Diana H. Wall, Ian D. Hogg

**Affiliations:** 1School of Science, University of Waikato, Hamilton 3240, New Zealand; 2LOEWE Centre for Translational Biodiversity Genomics, Senckenberg Biodiversity and Climate Research Centre, 60325 Frankfurt am Main, Germany; 3Canadian National Collection of Insects Arachnids and Nematodes, Agriculture and Agrifood Canada, Ottawa, ON K1A 0C6, Canada; 4British Antarctic Survey, Cambridge CB3 OET, UK; 5Department of Zoology, University of Johannesburg, Auckland Park 2006, South Africa; 6Millennium Institute Biodiversity of Antarctic and Subantarctic Ecosystems, BASE, University Austral of Chile, Valdivia 5090000, Chile; 7Securing Antarctica’s Environmental Future, School of Biological Sciences, Monash University, Melbourne 3800, Australia; 8Department of Biology, Evolutionary Ecology Laboratories, Brigham Young University, Provo, UT 84602, USA; 9Monte L. Bean Museum, Brigham Young University, Provo, UT 84602, USA; 10Department of Biology, Colorado State University, Fort Collins, CO 80523, USA; 11School of Global Environmental Sustainability, Colorado State University, Fort Collins, CO 80523, USA; 12Canadian High Arctic Research Station, Polar Knowledge Canada, Cambridge Bay, NU X0B 0C0, Canada

**Keywords:** speciation, geographic isolation, Acari, Antarctic conservation, DNA barcoding

## Abstract

Free-living terrestrial mites (Acari) have persisted through numerous glacial cycles in Antarctica. Very little is known, however, of their genetic diversity and distribution, particularly within the Ross Sea region. To redress this gap, we sampled mites throughout the Ross Sea region, East Antarctica, including Victoria Land and the Queen Maud Mountains (QMM), covering a latitudinal range of 72–85 °S, as well as Lauft Island near Mt. Siple (73 °S) in West Antarctica and Macquarie Island (54^o^S) in the sub-Antarctic. We assessed genetic diversity using mitochondrial cytochrome *c* oxidase subunit I gene sequences (COI-5P DNA barcode region), and also morphologically identified voucher specimens. We obtained 130 sequences representing four genera: *Nanorchestes* (*n* = 30 sequences), *Stereotydeus* (*n* = 46), *Coccorhagidia* (*n* = 18) and *Eupodes* (*n* = 36). Tree-based analyses (maximum likelihood) revealed 13 genetic clusters, representing as many as 23 putative species indicated by barcode index numbers (BINs) from the Barcode of Life Datasystems (BOLD) database. We found evidence for geographically-isolated cryptic species, e.g., within *Stereotydeus belli* and *S. punctatus*, as well as unique genetic groups occurring in sympatry (e.g., *Nanorchestes* spp. in QMM). Collectively, these data confirm high genetic divergence as a consequence of geographic isolation over evolutionary timescales. From a conservation perspective, additional targeted sampling of understudied areas in the Ross Sea region should be prioritised, as further diversity is likely to be found in these short-range endemic mites.

## 1. Introduction

Antarctic terrestrial arthropods have persisted for millions of years in microhabitats that are ice-free yet have some liquid water available [1,2,3]. The extent of these microhabitats, and how they are distributed across the landscape, is largely determined by the underlying geography and millennia-scale ice sheet dynamics [4], although across much of Antarctica the precise locations of refugia have not been identified. Even during periods of extensive ice cover, such as during the Last Glacial Maximum (LGM), in some regions, species likely survived either on protruding nunataks [1] or in association with geothermal features [5], although such features do not appear to explain the persistence of most terrestrial arthropod diversity, which is limited to lower-altitude coastal areas [1]. Long-term isolation of arthropod populations across the landscape, with resulting genetic bottlenecks, can ultimately lead to speciation through divergence via genetic drift or mutation [6]. However, as these isolated habitats in Antarctica are often difficult to access and sample, many locally unique genetic lineages are likely to await discovery. Mites are the most taxonomically diverse and widespread microarthropods in Antarctica, with at least 19 species currently recognized from the Ross Sea region alone [7,8,9] (Appendix A). However, estimates of their genetic diversity and levels of divergence among populations remain largely unknown.

Many Antarctic taxa are short-range endemics [10], limited to spatially small geographical areas within the continent. The logistical constraints inherent to remote Antarctic fieldwork, which have limited sampling of spatially isolated habitats, also limit the number of species that can be addressed in a single study. This is unfortunate, because taxonomic breadth is essential for gaining a more comprehensive knowledge of diversity across the landscape and for understanding processes of evolution and speciation. The ice-free terrestrial habitats of continental Antarctica have been classified into several biologically distinct regions (bioregions), known as Antarctic Conservation Biogeographic Regions (ACBRs) [11,12]. Within these bioregions there are also many smaller-scale landscape barriers to dispersal, further structuring the distribution of species or genetic lineages [13,14]. Studies using DNA-based methods for discriminating species, including cryptic species, have now been carried out for several Antarctic terrestrial invertebrate groups including rotifers [15], tardigrades [16,17], mites [18,19] and Collembola [20,21,22,23]. However, many of these studies are limited to small spatial scales, covering single bioregions [24].

Studies that incorporate both morphological and genetic data have shown that some Antarctic taxa that were originally thought to be widespread single species often consist of multiple, distinct species. For example, the collembolan *Friesea grisea* Schäffer with its type locality in sub-Antarctic South Georgia was widely reported from both maritime and continental Antarctica [25,26]. Until recently, this was one of the only micro-arthropod species thought to occur in both the latter regions as well as the sub-Antarctic. However, driven by the application of molecular phylogenetic analyses and also supported by detailed micromorphological assessments, this species has now been redescribed as at least five species [21,22,27,28]. Available molecular tools for detecting potential species-level differences include barcode index numbers (BINs), which are based on a clustering algorithm [29] and integrated into the Barcode of Life Datasystems (BOLD) database. Unique BINs can provide evidence to support putative species identifications and can also be used to quantify diversity where morphology is unclear for understudied or morphologically cryptic taxa.

Genetic markers for species-level assessments include the 5’ region of the mitochondrial cytochrome *c* oxidase subunit I (COI-5P) gene [30], which is the most commonly used DNA-based marker, particularly for Antarctic taxa such as Collembola [23,31,32]. Unfortunately, this standard “DNA barcoding” region has seen limited sequencing success for mites, e.g., [33], and previous Antarctic studies of the prostigmatid genus *Stereotydeus* have used the COI-3P region as an alternative [18,23,34,35]. Collectively, these studies have provided evidence of local speciation, including two new species descriptions for *S. nunatakis* and *S. ineffabilis* in Victoria Land [36]. However, the difficulty of obtaining specimens from a wider range of remote and isolated locations has limited the geographic coverage of available studies. Further, the absence of COI-5P sequences for Antarctic mites has limited a more widespread comparison with other taxa in Antarctica and elsewhere. Obtaining COI-5P sequences from Antarctic mite taxa would facilitate comparison with existing databases such as the Barcode of Life Datasystems (BOLD) database [37], as well as assist in identifying potentially distinct species and population genetic structure.

One Antarctic mite taxon that would particularly benefit from further attention is *Nanorchestes* (Endeostigmata)*,* a widespread genus of small-bodied mites that are also found globally [38]. Although yet to be subjected to similar molecular studies, the species *Nanorchestes antarcticus* Strandtmann, 1963 (type locality: Observation Hill, Ross Island, Victoria Land) likely provides an analogous example. It is another species that was originally recorded from both the maritime and continental Antarctic [8]. However, Strandtmann [39] subsequently described material from the former region as two distinct species, *N. berryi* and *N. gressitti*. Four new species were also described from Macquarie Island [40], and four species from the Ross Sea region [9], all of which were previously referred to as *N. antarcticus*. Considering the widespread geographic distribution of this taxon and its acknowledged morphological variability, it is likely that further species-level divergences will be revealed by the application of modern molecular approaches.

Here, using integrated morphological and molecular approaches, we assessed the species diversity of terrestrial mites collected from the Ross Sea region of Antarctica, including North Victoria Land (NVL), South Victoria Land (SVL) and the Queen Maud Mountains (QMM), covering a regional latitudinal range of 72–85 °S. We also obtained mites from Lauft Island near Mt. Siple (73 °S) in West Antarctica and Macquarie Island (54 °S) in the sub-Antarctic. We used mitochondrial DNA cytochrome *c* oxidase subunit I gene sequences (COI-5P DNA barcode region) to assess levels of genetic diversity within and among locations, as well as assist with species-level identifications. Our data provide a framework for ongoing studies as well as initiate a reference library for the molecular identification (DNA barcoding) of Antarctic taxa.

## 2. Materials and Methods

### 2.1. Study Area and Specimen Collection

Sampling for free-living mites was undertaken during austral summer field seasons between 2008 and 2018 at 16 locations throughout the Ross Sea region (continental Antarctica), spanning latitudes from 72 to 85 °S, and occupying the longitudinal wedge between 160 °E and 160 °W (Figure 1). This region includes three biologically distinct zones, or Antarctic Biogeographic Conservation Regions (ACBRs) *sensu* [11,12], North Victoria Land (NVL), South Victoria Land (SVL) and the Queen Maud Mountains (QMM, part of the Transantarctic Mountains ACBR (TAM)). The Queen Maud Mountains represent some of the southern-most terrestrial ice-free habitats in Antarctica. Further samples were obtained in 2017 from the previously unsampled Lauft Island (unofficial name near Mt. Siple; 73 °S, 127 °W) in West Antarctica and sub-Antarctic Macquarie Island (54 °S, 159 °E). All specimens were collected using modified aspirators [41], pitfall traps [42] or flotation from soil samples [43] and immediately preserved in 100% ethanol.

### 2.2. Genetic and Morphological Analyses

For genetic analyses, we used DNA sequences from the mitochondrial cytochrome *c* oxidase subunit I gene (COI-5P DNA barcode region) [30]. On return of the specimens from Antarctica, DNA sequencing was undertaken either at the University of Waikato (UoW), New Zealand or the Canadian Centre for DNA barcoding (CCDB), Canada. For total DNA extraction, a REDExtract-NAmp kit (Sigma-Aldrich, Merck KGaA, Germany) was used at the UoW, while a glass fibre plate method (AcroPrep, Pall Corp., New York, NY, USA) was used at the CCDB. The COI gene was amplified using the universal primers HCO2198 and LCO1490 [44] at the UoW, and the combination of HCO2198, LCO1490 together with LepF1 and LepR1 [45] at the CCDB. At the UoW, PCR amplifications for each specimen were carried out in 15 µL volumes containing 7.5 µL PCR master mix solution (i-Taq) (Intron Biotechnology), 0.2 µM (0.3 µL) of each primer and 1 µL of DNA extract (unquantified). Thermal cycling conditions were: 94 °C for 5 min followed by 5 cycles (94 °C for 1 min, 48 °C for 1.5 min and 72 °C for 1 min), then 35 cycles (94 °C for 1 min, 52 °C for 1.5 min and 72 ^o^C for 1 min) of denaturation and polymerase amplification, with a final elongation at 72 °C for 5 min. At CCDB, thermal cycling conditions were: 94 °C for 1 min followed by 5 cycles (94 °C for 1 min, 45 °C for 1.5 min and 72 °C for 1.5 min) then 60 cycles (94 °C for 1 min, 50 °C for 1.5 min and 72 °C for 1 min) of denaturation and polymerase amplification, with a final elongation at 72 °C for 5 min. Successful amplification products were cleaned with 0.1 μL exonuclease (EXO) (10 U/μL) and 0.2 μL shrimp alkaline phosphate (SAP) (1 U/μL) (Illustra from GE Healthcare) at 37 °C for 30 min then 80 °C for 15 min at the UoW, or Sephadex at CCDB. Sequencing was carried out in forward and reverse directions using an ABI 3130 at the UoW or an ABI 3730x1 sequencer at the CCDB. Sequences were uploaded to the Barcode of Life Datasystems (BOLD) database. Specimen collection details, photographs, primers used and full sequence data are available from the BOLD database under dataset DS-ANTMIT (dx.doi.org/10.5883/DS-ANTMIT, accessed on 17 March 2023).

Following DNA extraction, the exoskeletons for some individual specimens were carefully removed from the extract solution and permanently slide-mounted in polyvinyl alcohol (PVA) mounting medium (BioQuip Products, Rancho Dominguez, CA, USA) for morphological assessment. The slide-mounted specimens were then identified with the aid of a compound microscope using the available taxonomic literature e.g., [8,9,40]. All mounted slides are housed at the Canadian National Collection of Insects, Arachnids and Nematodes (Agriculture and Agrifood Canada, Ottawa, ON Canada) and the remainder of the specimens are housed at the Centre for Biodiversity Genomics (University of Guelph, Guelph, ON, Canada). Identifications were possible for 12 specimens representing five currently recognized species. For any uncertainties with species-level identifications, the more conservative genus-level assignment was used.

### 2.3. Data Analyses

In addition to our newly generated sequences, 491 publicly available COI-5P sequences of at least 400 bp in length were downloaded from GenBank in July 2022 for the genera *Nanorchestes*, *Eupodes* and *Rhagidia*. No previous sequences were available for *Stereotydeus* or *Coccorhagidia*. None of the existing public records were from Antarctica or the sub-Antarctic. To simplify the analyses, we reduced these public records to one sequence per unique barcode identification number (BIN) or named species (*n* = 26). The final sequence alignment of 156 sequences, each ranging from 404 to 658 bp in length, included our new sequences (*n* = 130) as well as the 26 representative publicly available sequences (*n* = 11 *Eupodes*, *n* = 10 *Nanorchestes*, *n* = 5 *Rhagidia*). All sequences were aligned in Geneious Prime v2021.2.2 using MUSCLE 3.8.425 and then exported as a PHYLIP file for downstream analyses. A maximum likelihood tree was built using IQ-TREE 2 [46] (-nt AUTO -s alignment.phy -st DNA -m MFP -bb 1000 -bnni -alrt 1000) where the best fit model of TIM+F+R4 chosen according to BIC was inherently applied during the tree-building process. The resulting tree file was then imported to R using read.tree from ggtree v3.1.5.900 [47] for visualisation, along with tidyverse v1.3.1 [48] and treeio v1.17.2 [49]. Appendix A provides collection details, BINs and BOLD codes for each sequence included in the phylogenetic tree. Haplotype networks were created in R for *Stereotydeus* and *Nanorchestes*, using pegas v1.1 [50] and phylotools v0.2.2 [51], and are available as Appendix A).

Based on the maximum-likelihood phylogenetic tree, monophyletic clades were considered as “genetic clusters”, or putative species, if members of the group had already been taxonomically identified to species level (slide-mounted specimens) or otherwise based on location, as there was a clear genetic distinction between regions. A barcode gap analysis [52] was then performed on these putative species groups using ape v5.6.2 [53], spider v1.5.0 [54] and ggplot2 v3.3.5 [55]. Following the genetic analyses, all specimens that occurred within the same BIN as the morphologically identified specimens were also attributed to this taxonomic name following protocols in deWaard et al. [56].

## 3. Results

We obtained 130 COI-5P sequences for individual free-living mites collected from 16 locations throughout the Ross Sea region as well as from the Lauft and Macquarie Islands. Each specimen was identified morphologically to at least the genus level (Prostigmata: *Coccorhagidia, Eupodes, Stereotydeus*; Endeostigmata: *Nanorchestes*), and five taxa (12 specimens) were identified to species level (*Coccorhagidia gressitti*, *C. keithi*, *Eupodes wisei*, *Stereotydeus belli* and *S. punctatus)*. A maximum likelihood tree delineated the Antarctic sequences into 13 major monophyletic groups (genetic clusters), 5 of which represented the taxonomically identified species, and none of which contained the publicly available (i.e., non-Antarctic) COI-5P sequences (Figure 2 and Appendix A). The 13 clusters were each separated by 8–19% sequence divergence (Figure 3; Table 1). Seven of the clusters were represented by a single barcode index number (BIN) and contained < 2.9% sequence divergence, while the remaining six clusters each contained 2–3 BINs and 3.7–14% sequence divergence (Table 1).

Our genetic data support the taxonomic species designations of *C. keithi* and *E. wisei*, as they were represented by a single BIN each. However, we found multiple BINs (i.e., potential cryptic species) for *C. gressitti, S. belli* and *S. punctatus* (Table 1). The three BINs of *C. gressitti* (>6% divergence) were all present at Christie Peak, while the BINs for *S. belli* and *S. punctatus* were geographically isolated. *Stereotydeus belli* and *S. punctatus* each had a unique population found only at Cape Hallett, which was genetically distinct from their respective populations found either at Redcastle Ridge (*S. belli*; 7.8% divergent) or Tombstone Hill (*S. punctatus*; 11.04% divergent).

Three further genetic clusters (*Nanorchestes* sp. 1 and sp. 6, *Stereotydeus* sp. 4), which were identified morphologically to genus level, also included more than one unique BIN. These BINs were distributed differently across the landscape, depending on the taxon. For instance, some taxa had all BINs present at a single location, while for other taxa unique BINs were each found at a different location. All three BINs of *Nanorchestes sp.* 1 (>6% divergence) were found at Ebony Ridge, and the two BINs for *Nanorchestes* sp. 6 (>3% divergence) were found at the same site on Macquarie Island. In contrast, BINs were unique to single locations for three of the four *Stereotydeus* clusters, the exception being Cape Hallett, where two BINs for *S. punctatus* were present in sympatry (Table 1).

Based on the morphological and genetic data, we identified two locations at which four or more putative mite species were present (Table 1). At Cape Hallett in North Victoria Land (72 °S), *C. gressitti*, *E. wisei*, *S. belli* and *S. punctatus* were present in sympatry. Two unique BINs for *S. punctatus* were also found at Cape Hallett, giving a total of five unique mite BINs at this location. At Mount Franke in the Queen Maud Mountain region (84 °S), two genetic clusters each of *Stereotydeus* and *Nanorchestes* were present (five unique BINs in total), representing 4–5 putative species.

## 4. Discussion

We assessed the genetic diversity of Antarctic mites from across a wide geographic and taxonomic range in the Ross Sea region. We obtained 130 standard COI-5P DNA barcode sequences *sensu* [30] from individual mites representing the four genera *Nanorchestes, Stereotydeus, Coccorhagidia* and *Eupodes.* The sequences were grouped into 13 distinct genetic clusters and a total of 23 unique BINs, some of which were found at only a single location. We also identified locations where multiple clusters or BINs occurred in sympatry, highlighting the potential conservation priorities for protecting sites with high levels of diversity, in keeping with the Antarctic Specially Protected Area priorities outlined by the Protocol on Environmental Protection to the Antarctic Treaty [57].

*Nanorchestes* was the most genetically diverse and widespread genus in our dataset, with six major genetic clusters distributed across ten locations, covering all three ACBRs in the Ross Sea region [11,12] as well as Lauft Island and Macquarie Island. Although considerable morphological variation was noted in the original description of *Nanorchestes antarcticus* [8], this taxon was subsequently divided into multiple distinct species based on detailed morphology [9,39,40]. Based on the geographic range of each of the genetic clusters we found, it is likely that *Nanorchestes* sp. 3 in NVL is either *N. antarcticus* or *N. lalae*, *Nanorchestes* sp. 5 in SVL could represent either *N. bellus, N. lalae* or *N. antarcticus*, and *Nanorchestes* sp. 1 and 4 in the QMM could represent *N. lalae* and *N. brekkeristae*. We also found a genetically distinct cluster (*Nanorchestes* sp. 2) at Lauft Island near Mt. Siple, a location from where microarthropods including *Nanorchestes* had not previously been sampled or described. The individuals collected from Lauft Island were genetically most similar to the *Nanorchestes* found in the QMM region (8% sequence divergence), and we speculate that refugial locations in the QMM may have potentially acted as a source habitat for Lauft Island. Based on available physical modelling and biological evidence [58,59] areas in the QMM were likely to have remained ice-free during the last glacial maximum (LGM). In the absence of the present-day Ross Ice Shelf during warmer periods [59], individuals could have dispersed (e.g., via rafting on ocean currents) from the QMM and populated any suitable habitats near Lauft Island until the Ross ice sheet reformed [60]. Future morphological and multi-gene analyses focused on individuals from both Lauft Island and the QMM would help distinguish putative species and the evolutionary relationships between the *Nanorchestes* populations at these two locations.

*Stereotydeus belli* and *S. punctatus* are currently known only from the North Victoria Land region, with *S. punctatus* reported from only a single location, Crater Cirque [34]. However, we found genetic variants of *S. punctatus* (three BINs) from two additional locations in NVL, suggesting that even further genetic diversity may exist within this species. *Stereotydeus belli* and *S. punctatus* were each represented by multiple BINs and some of these BINs were geographically isolated. Both species had a unique population found only at Cape Hallett and a genetically divergent population (7–11% sequence divergence) at either Tombstone Hill or Redcastle Ridge. Both Redcastle Ridge and Tombstone Hill are located towards the inner end of the Edisto Inlet, presumably isolated from Cape Hallett by local marine currents. Indeed, the isolation of Cape Hallett from other NVL sites has already been recognised for *S. belli* [34].

Patterns of genetic variability among sites in the QMM differed depending on the taxon considered. We found two genetic clusters of *Stereotydeus* in the QMM and it is plausible that at least one of these is *S. shoupi*, the only *Stereotydeus* species currently described and recorded from this region [8]. Demetras et al. [18] previously addressed the genetic diversity of *S. shoupi* from the QMM using the alternative COI-3P gene region and found 8% genetic divergence between the Darwin Glacier (79 °S) and Beardmore Glacier (83 °S). We also found high levels of genetic divergence in this region, of up to 14% between populations at Beardmore Glacier and the more southerly Shackleton Glacier (84 °S), for *Stereotydeus* sp. 4 (which also could be *S. shoupi*). Such high levels of genetic variation between the Beardmore and Shackleton Glaciers have also been found for Collembola, with over 13% COI-5P sequence divergence in *Antarctophorus subpolaris* between these two locations [58]. In contrast, *Nanorchestes* had shared BINs between the Beardmore and Shackleton Glacier sites. One possible explanation is that a smaller body size and greater tolerance of lower environmental temperatures for *Nanorchestes* relative to *Stereotydeus* [61] facilitated increased levels of dispersal among habitats. However, further work *sensu* [42,62,63] would be required to test this hypothesis.

*Stereotydeus* sp. 3 was found only in the vicinity of the Shackleton Glacier, and could represent a more southern population for a northern species of *Stereotydeus.* This has previously been found for the collembolan *Gomphiocephalus hodgsoni*, a common SVL species that was only recently reported from the QMM region [58]. Alternatively, *Stereotydeus* sp. 3 may represent a previously undescribed species unique to the area. Similar to *Stereotydeus* sp. 3, *Tullbergia mediantarctica* (Collembola) is also only known from a small area near the mouth of the Shackleton Glacier in the QMM and has previously been identified as potentially vulnerable to extinction due to this narrow distributional range [58]. We suggest that some of the mite taxa we have identified with narrow geographic distributions may also be vulnerable under future climate scenarios [64].

Four genetic clusters were found at Mount Franke, near the Shackleton glacier in the QMM region, probably representing two species of *Stereotydeus* and two of *Nanorchestes*. While reductions in biodiversity are often observed with increasing latitude, we found no evidence of this occurring in the Ross Sea region, as a similar level of diversity was found at both QMM and NVL (four genetic clusters each). Caruso et al. [65] and Colesie et al. [66] also concluded that levels of diversity were more likely to have been influenced by microhabitat water availability rather than macroclimatic temperature changes associated with latitude in this region. For example, we found six *Nanorchestes* species clusters, each with different distributional ranges, which could be due to different habitat preferences, a feature already hypothesised among other species of this genus from the Antarctic Peninsula region [67]. Notably, some soils that were exposed during the LGM may provide less suitable habitats today owing to salt accumulation [68,69,70]. While abiotic factors are usually considered the predominant factor shaping biological communities across continental Antarctica [71], it is possible that additional biotic interactions have remained undetected [72,73].

For the five species that were taxonomically verified, we found 0.46% to 11.04% sequence divergence among populations. Previous genetic studies of Antarctic mites have suggested that they may have differing levels of genetic divergence relative to sympatric Collembola. Possible explanations include ecological factors, such as mites being able to colonize and tolerate newly available habitats before Collembola (and hence longer potential divergence time) or that they have differing rates of evolution [18,23,35]. The Dry Valleys are the most accessible and widely studied ice-free areas in the Ross Sea region. However, at the time of those studies, only *Stereotydeus mollis* was known from this area. With the recent description of an additional species in the same area, *S. ineffabilis* [36], we suggest the possibility that the “intraspecific diversity” described for *S. mollis* by McGaughran et al. [35] and Stevens and Hogg [23] may be the result of genetic differences between *S. mollis* and *S. ineffabilis*. Furthermore, all existing mite studies that include DNA-based methods [18,23,34,35] have used the alternative COI-3P gene region. The COI-3P gene region is more suitable for resolving more deeply-rooted phylogenetic relationships and is more reliably amplified and sequenced than the standard COI-5P region, which has a particularly low success rate (<50%) in mites [74]. Unfortunately, direct comparisons of genetic divergence for mites and Collembola are only likely to be accurate when the same gene region is used. The COI-5P sequences we provide here can now be used as a reference for future comparisons of divergence rates.

In conclusion, based on sequencing of the COI-5P gene region, we found between 13 and as many as 23 putative species representing four mite genera, far exceeding previously recognized diversity from this geographic region. Further mite diversity is likely to remain undetected and ongoing and targeted sampling initiatives, e.g., [75] will advance this field. Understudied areas that should be specifically targeted include the area south of the Drygalski Ice Tongue (76 °S) in NVL, around the Darwin Glacier (80 °S) in the Transantarctic Mountains and in the vicinity of the Byrd Glacier (82 °S) in the QMM. We highlight the value of DNA barcoding for providing initial evidence of species-level diversity, and encourage combined morphological and molecular approaches as part of a more rigorous assessment of distinct species. The standardized DNA sequences (barcodes) we provide serve as a baseline reference library for mites in the Antarctic and will facilitate future DNA-based studies, including comparative studies among locations and taxa. Future studies are likely to benefit from multi-gene or meta-genomic approaches. Depending on the quality of DNA preservation, future studies may be able to use existing sample collections and limit the need for repeat visits to previously visited locations, thereby helping to reduce the costs and carbon footprint of conducting research in Antarctica [76]. However, ongoing fieldwork will be required for fresh specimens (non-degraded) that may not have yet been analysed for DNA variability (e.g., *Stereotydeus villosus* from the maritime Antarctic) or for targeting specific taxa that may not yet have been collected, such as the oribatid mite *Maudheimia petronia* from NVL or *Protereunetes maudae* (Eupodidae) from QMM [8].

Overall, our data suggest that the combination of highly isolated locations and long evolutionary timescales have resulted in high levels of genetic differentiation and local speciation for mites throughout the Ross Sea region. A fuller understanding of the locally unique and endemic mite taxa of Antarctica, which occupy the most southern terrestrial habitats on Earth, will provide valuable insights into the evolutionary history of the Antarctic landscape.

## Figures and Tables

**Figure 1 genes-14-00606-f001:**
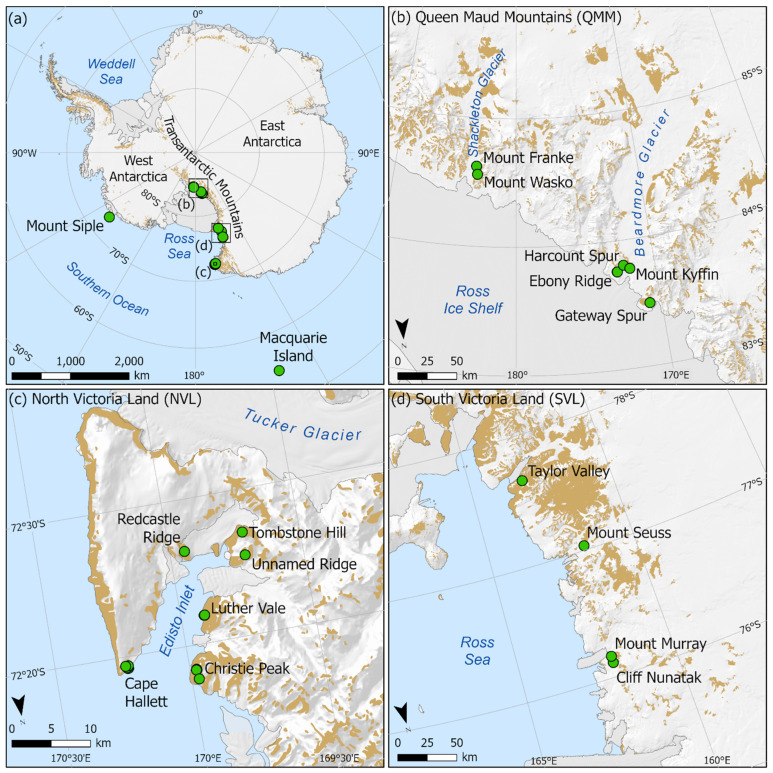
Standard COI barcodes were obtained from 130 individual mites that were collected from 16 sampling locations throughout the Ross Sea region and Lauft Island and Macquarie Island (**a**), including the Queen Maud Mountains (**b**) and Victoria Land (**c**,**d**).

**Figure 2 genes-14-00606-f002:**
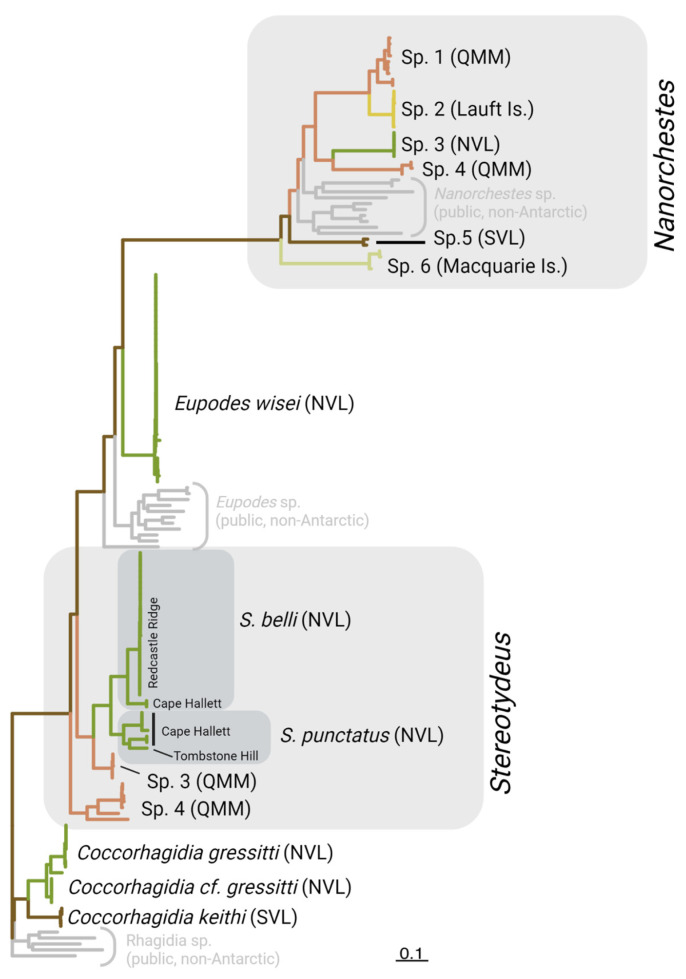
Maximum likelihood tree including 26 publicly available COI-5P mite sequences (in grey), as well as our 130 individual mites sequenced from the Ross Sea region that were grouped into 13 genetic clusters (putative species) based on morphological identifications and sequence-based analyses. Each cluster was found in only one of the major biogeographic regions, being restricted to one of the Queen Maud Mountains (QMM), South Victoria Land (SVL) or North Victoria Land (NVL). The Lauft Island population of *Nanorchestes* was most similar to one of the QMM species. See the Appendix A for a more detailed version of the tree. Created with BioRender.com.

**Figure 3 genes-14-00606-f003:**
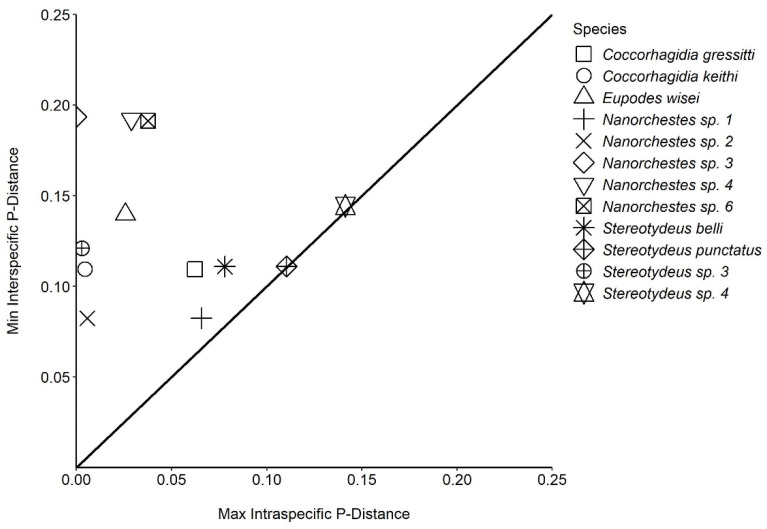
Barcode gap analysis confirmed that our genetic-based clustering into putative species-level groups was appropriate and in agreement with species named in traditional morphological taxonomy (for those specimens that could be identified to species level using available taxonomic keys). See Table 1 and Appendix A for precise intraspecific and interspecific divergence values.

**Table 1 genes-14-00606-t001:** Geographic distribution of the 13 genetic clusters of Antarctic mites identified in this study; NVL: North Victoria Land, SVL: South Victoria Land, QMM: Queen Maud Mountains, (B): in the vicinity of Beardmore Glacier, (S): Shackleton Glacier.

Genus		*Coccorhagidia*		*Eupodes*	*Nanorchestes*						*Stereotydeus*				Totals
Putative Species Clusters		*gressitti*	*keithi*	*wisei*	sp. 1	sp. 2	sp. 3	sp. 4	sp. 5	sp. 6	*belli*	*punctatus*	sp. 3	sp. 4	13
n sequences		14	4	36	9	7	5	3	2	4	27	7	5	7	130
	n BINs	3	1	1	3	1	1	1	1	2	2	3	1	3	23
	Max. intra.	6.23%	0.46%	2.58%	6.56%	0.57%	0.00%	2.90%	NA	3.77%	7.80%	11.04%	0.30%	14.13%	
	Min. inter.	10.94%	10.94%	13.97%	8.24%	8.24%	19.35%	19.21%	NA	19.13%	11.09%	11.09%	12.10%	14.44%	
Region	Location	**Locations where each putative species cluster was found**													**Total n clusters per location**
	Macquarie Is.									[X] **					
	Lauft Island					X *									
NVL	Cape Hallett	X		X							X *	X **			4
NVL	Christie Peak	[X]		X											2
NVL	Luther Vale	X		X											2
NVL	Unnamed Ridge	X													1
NVL	Tombstone Hill						X *					X *			2
NVL	Redcastle Ridge										X *				1
SVL	Cliff Nunatak		X												1
SVL	Mt. Murray		X												1
SVL	Mt. Seuss								X						1
SVL	Taylor Valley								X						1
QMM (B)	Gateway Spur													X *	1
QMM (B)	Ebony Ridge				[X]			X							2
QMM (B)	Harcourt Spur				X										1
QMM (B)	Mt. Kyffin				X										1
QMM (S)	Mt. Franke				X			X					X	X *	4
QMM (S)	Mt. Wasko							X					X	X *	3
Total n locations		4	2	3	4	1	1	3	2	1	2	2	2	3	

[X]: all BINs for this species cluster were found in sympatry at this location; X *: this location harboured a BIN that was found nowhere else; X **: two BINs were found here and nowhere else.

## Data Availability

All original photographs, specimen and sequence data are available online from the Barcode of Life DataSystems (BOLD) database (www.boldsystems.org (accessed on 26 February 2023)) under dataset DS-ANTMIT (dx.doi.org/10.5883/DS-ANTMIT (accessed on 17 March 2023)).

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
