# Peer review of "Biogeography and Genetic Diversity of Terrestrial Mites in the Ross Sea Region, Antarctica"

_genes, 2023, doi:10.3390/genes14030606_

Round 1
Reviewer 1 Report
The manuscript reports on a DNA barcoding study from arthropod samples collected from remote sites in the Ross Sea region of Antarctica. The sampling sites cover a vast area and a large timespan of collection (10 years). As such, this study represents an incredibly valuable resource in understanding the biodiversity of an extreme and hugely understudied environment. While this work is largely descriptive in nature, the effort and thoroughness applied to the study cannot be understated. The authors have made an incredible attempt to make their findings relevant by sequencing a difficult region of mtDNA for mites but one that aligns the results with most other studies of arthropods. This makes comparisons of biodiversity much more meaningful for this dataset. The results are concise and clearly presented, making for a manuscript which is readily interpreted and accessible for most researchers of arthropod diversity. I commend the authors on an interesting and comprehensive study.
The only concern with this manuscript is one of terminology. The authors jump right into technical barcoding language with little explanation for the reader. It is important, particularly for readers not familiar with barcoding terms, to clearly define what is meant by the various taxonomic categories referred to in the text. Terms such as BIN (line 171 and elsewhere), cluster (line 219 and elsewhere), and putative species (line 202 and elsewhere) should be clearly explained in the introduction. It would also be worthwhile to spend some time in the discussion to explore the accepted taxonomic process for how samples with divergent COI sequences generate enough support to be accepted as a distinct species. Such an explanation will help acknowledge some of the controversy around the use of barcoding sequence data alone in taxonomy and systematics and further highlight the significance of this data in a purely biodiversity and conservation context.
Author Response
Reviewer Point 1: The only concern with this manuscript is one of terminology. The authors jump right into technical barcoding language with little explanation for the reader. It is important, particularly for readers not familiar with barcoding terms, to clearly define what is meant by the various taxonomic categories referred to in the text. Terms such as BIN (line 171 and elsewhere), cluster (line 219 and elsewhere), and putative species (line 202 and elsewhere) should be clearly explained in the introduction.
Author Response to Point 1: We have made the following changes throughout the manuscript to address this valuable comment:
Lines 82-87: We have now added “Available molecular tools for detecting potential species-level differences include Barcode Index Numbers (BINs) which are based on clustering algorithms [29] and are integrated into the Barcode of Life Datasystems (BOLD) database. Unique BINs can provide evidence to support putative species identifications and can also be used to quantify diversity where morphology is unclear, for understudied or morphologically cryptic taxa.”
Lines 166-169: We have clarified this sentence to remove mention of confusing terminology.
Line 184: We have added further explanation to the sentence to now read “To simplify the analyses, we reduced these public records to one sequence per unique Barcode Identification Number (BIN) or named species (n = 26).”
Lines 198-202: We added to the methods “Based on the maximum likelihood phylogenetic tree, monophyletic clades were considered as “genetic clusters”, or putative species, if members of the group had already been taxonomically identified to species-level (slide-mounted specimens) or otherwise based on location, as there was a clear genetic distinction between regions.”
Line 218: We have revised “genetically-distinct clusters” to “major monophyletic groups (genetic clusters)”
Point 2: It would also be worthwhile to spend some time in the discussion to explore the accepted taxonomic process for how samples with divergent COI sequences generate enough support to be accepted as a distinct species. Such an explanation will help acknowledge some of the controversy around the use of barcoding sequence data alone in taxonomy and systematics and further highlight the significance of this data in a purely biodiversity and conservation context.
Response to Point 2: We have now added the following statement (Lines 379-382): “We highlight the value of DNA barcoding for providing initial evidence of species-level diversity, and encourage combined morphological and molecular approaches as part of a more rigorous assessment of distinct species.”
Reviewer 2 Report
The article is well written and the results were discussed using an extensive bibliography, emphasizing previous studies in the Antarctic region. I did not find any mistakes, in my vision this article can be published in the current version.
Author Response
Reviewer comment: ..in my vision this article can be published in the current version.
Author response: We very much appreciate this positive support for the manuscript.